# Antibiofilm Effect of Silver Nanoparticles in Changing the Biofilm-Related Gene Expression of *Staphylococcus epidermidis*

**DOI:** 10.3390/ijms23169257

**Published:** 2022-08-17

**Authors:** Denis Swolana, Małgorzata Kępa, Celina Kruszniewska-Rajs, Robert D. Wojtyczka

**Affiliations:** 1Department of Microbiology and Virology, Faculty of Pharmaceutical Sciences in Sosnowiec, Medical University of Silesia, ul. Jagiellońska 4, 41-200 Sosnowiec, Poland; 2Department of Molecular Biology, Faculty of Pharmaceutical Sciences in Sosnowiec, Medical University of Silesia in Katowice, ul. Jedności 8, 41-200 Sosnowiec, Poland

**Keywords:** antimicrobial activity, biofilm, nanosilver, *Staphylococcus epidermidis*

## Abstract

Nowadays, antibiotic resistance is a major public health problem. Among staphylococci, infections caused by *Staphylococcus epidermidis* (*S. epidermidis*) are frequent and difficult to eradicate. This is due to its ability to form biofilm. Among the antibiotic substances, nanosilver is of particular interest. Based on this information, we decided to investigate the effect of nanosilver on the viability, biofilm formation and gene expression of the *icaADBC* operon and the *icaR* gene for biofilm and non-biofilm *S. epidermidis* strains. As we observed, the viability of all the tested strains decreased with the use of nanosilver at a concentration of 5 µg/mL. The ability to form biofilm also decreased with the use of nanosilver at a concentration of 3 µg/mL. Genetic expression of the *icaADBC* operon and the *icaR* gene varied depending on the ability of the strain to form biofilm. Low concentrations of nanosilver may cause increased biofilm production, however no such effect was observed with high concentrations. This confirms that the use of nanoparticles at an appropriately high dose in any future therapy is of utmost importance. Data from our publication confirm the antibacterial and antibiotic properties of nanosilver. This effect was observed phenotypically and also by levels of gene expression.

## 1. Introduction

*Staphylococcus epidermidis* (*S. epidermidis*) is one of the most common Coagulase-Negative Staphylococci (CoNS) [1]. The average person carries 10 to 24 strains of this bacteria on the surface of the skin and mucous membranes. At the same time, this pathogen is the most common etiological factor of implant-related infections [2]. The cause of these common opportunistic infections is biofilm. As the most important pathogenicity factor of this staphylococcus, it plays a key role during infection [3,4].

Biofilm is a community of microorganisms which lives on a specific surface and is composed of cells and the extracellular matrix, which is made up of exopolysaccharides (EPS), proteins and extracellular DNA (eDNA). Infections caused by biofilm-producing bacteria are difficult to treat due to their greater resistance to host immune defense mechanisms and the appearance of genotypic mutations that alter drug targets [5,6]. Most authors divide the process of creating this multicellular structure into three main stages: adhesion, maturation and dispersion [7]. The stages of biofilm formation are shown in Figure 1.

### Biofilm Generation Process

At the time of exposure to stress, bacterial cells in the host organism face the possibility of being destroyed by the host’s innate immune response and can transform from an “active” form to a “passive”, “adherent” form. This is favored by surface conditions—microorganisms can adhere to surfaces with different adhesion (inert or biotic). Bacterial cells can reach such sites by chemotaxis. Then, they produce adhesins, which play a key role in the process of biofilm formation, which do not appear on the surface of planktonic cells, as well as surface molecules, such as protein surface autolysins or teichoic acids, which change the physicochemical properties of the bacterial cell surface. Staphylococci produce Microbial Surface Components Recognizing Adhesive Matrix Molecules (MSCRAMMs), which determine covalent and non-covalent bonding to the surface where the biofilm is to be formed. When a stimulating factor occurs, the physiology of bacteria rapidly changes and is oriented towards the formation of a bacterial community [8].

As soon as the cells adhere to the surface, the maturation phase takes place, during which the production of bacterial mediators increases, metabolism slows down and intercellular cooperation is emphasized. Bacteria aggregate to form the extracellular matrix and become more virulent due to an increased resistance to antibiotics and the host’s immune system [3]. The largest part of the extracellular matrix is formed by exopolysaccharides, among which the main role is played by polysaccharide intercellular adhesin (PIA), or poly-β-N-acetylglucosamine. Structurally, it is a β-1,6 linkage partially deacylated N-acetylglucosamine polymer, which, along with other polymers such as teichoic acids and proteins, forms the basis of the mucus. Due to its structure, it is an electropositive molecule, which promotes electrostatic interactions between the negatively charged surfaces of adjacent bacteria, acting as adhesives. The PIA adhesin is encoded by the *ica* gene, which is located in the *icaADBC* operon. Its activity is regulated by environmental factors (e.g., oxygen concentration, glucose, ethanol, osmolarity, temperature and antibiotics, such as tetracycline) as well as autogenous factors (e.g., SarA and SigB proteins which enhance PIA expression, whereas LuxS protein slows down PIA expression) [8]. The arrangement of the *ica* operon in *S. epidermidis* can be found in Figure 2. The *icaA* and *icaD* genes are closely involved in exopolysaccharide synthesis. Similarly, *icaB* and *icaC* participate in the synthesis of the poly-N-acetylglucosamine polymer through its translocation to the bacterial cell surface and deacetylation of exopolysaccharide molecules [9,10]. IcaA N-acetylglucosaminyltransferase, with the help of icaD, acts on the substrate UDP-N-acetylglucosamine and forms an exopolysaccharide. The reaction product is then translocated across the cytoplasmic membrane by icaC. In turn, deacetylation of PIA by icaB allows PIA to adhere to biotic and abiotic surfaces, facilitating biofilm formation [3]. Although PIA adhesin plays a key role in the biofilm synthesis process, it is not necessary for biofilm production—there are biofilm-forming strains which do not contain any *ica* operon genes [8].

In the PIA-dependent model of biofilm production, polysaccharide intercellular adhesin is formed from the translation products of genes contained in the *ica* locus— *icaADBC* operon, which is responsible for biosynthesis, and the *icaR* gene, which is related to regulation. In the anaerobic environment of biofilm, expression of these genes is enhanced, inter alia, by the SrrAB (staphylococcal respiratory response regulator) protein [7,8,12,13]. PIA expression can be inhibited by the transcriptional regulator TcaR (teicoplanin-associated locus regulator), Spx (a global regulator of the stress response genes) and IcaR protein, as well as by integration into the genes of the *ica* operon in the IS25 insertion element [14]. Upstream and divergently transcribed *icaR*, which belongs to the *TetR* (tetracycline repressor) family, is involved in gene regulation, acting as either transcriptional activators or repressors. Like other members of this family, it regulates transcription by binding to the promoter region. This gene encodes the transcription of the *ica* operon repressor and works by binding to a specific DNA sequence directly before *icaA* [15]. Structural studies have shown that IcaR binds as two homodimers to DNA and does not regulate its own expression. Sigma factor σ^B^ has been shown to indirectly repress *icaR* transcription. On the other hand, TcaR can repress both *icaADBC* and *icaR* transcription [16]. However, the Rbf (regulator of biofilm formation) protein enhances *ica* gene expression and the production of PIA by inhibiting *icaR* transcription [7,8,12,13].

In the case of *S. epidermidis*, the communication system between staphylococcal biofilm cells is based on the phenomenon of quorum sensing. This system consists of many molecules and can inhibit and activate production of the biofilm structure. The three-dimensional structures created in this way take various shapes, with water channels inside them, ensuring the supply of food to cells located in the deeper layers of the community and allowing the possibility of transmitting intercellular signals [7,13]. In the final stage of biofilm development, dispersion occurs, i.e., cells detach from the surface of the structure. In good environmental conditions, the biofilm is expanded by blood or other body fluids and other places are colonized by detached cells [17,18,19].

The biofilm, for which the development process is described above, occurs in *S. epidermidis* and makes it difficult for clinicians to treat infections associated with implants in the human body. Low sensitivity to conventional antimicrobials has increased the need for new strategies to combat bacteria with this growth pattern. Attention has been paid to substances of a natural origin with antimicrobial properties. Furthermore, modifications have been made to the type of material used for implantation in order to prevent infection. There have also been attempts to use antibiotics immobilized on the surface of biomaterials. Finally, combinations of antibiotics with other antimicrobials have been tested in order to reduce the likelihood of resistance and enhance the synergistic effect of individual substances [20,21].

The antibacterial activity of nanosilver is well known. Silver nanoparticles (AgNPs) show antibacterial and antibiofilm properties through the lysis of cell membranes, generation of reactive oxygen species (ROS), destabilization of ribosomes, mitochondria, DNA and RNA or through the disruption of cell pathways [22,23,24]. The mechanism of their action against staphylococci involves the irreversible damage of bacterial cells by inhibiting bacterial DNA replication, the degradation of the bacterial cytoplasm membranes or modification of intracellular adenosine-5’-triphosphate (ATP) levels. They can damage the cell membrane through direct contact and attack the respiratory chain. They can also damage the peptidoglycan in the cell wall of Gram-positive and Gram-negative bacteria. Additionally, AgNPs also damage lipoteichoic acids present in the cell wall of Gram-positive bacteria. In the presence of nanosilver particles, superoxide and hydroxyl radicals are formed without the participation of cells. It has been shown that these radicals damage intracellular molecules [25]. For this reason, nanosilver has been used, among others, in medicine in addition to biomaterials [26]. It has been suggested that the size, shape, concentration and even the production method of the nanoparticles influence their antibacterial capacity [22].

Nanotechnology can be used as a weapon against multi-drug resistant bacteria. Nanoparticles are small substances ranging in size from 1 to 100 nm. Silver nanoparticles have attracted the attention of researchers in recent years. Their anti-biofilm activity is also very promising [23,26,27]. This article shows the effect of silver nanoparticles with a diameter of 10 nm at different concentrations on the viability and the ability to produce biofilm by *S. epidermidis* bacteria. Moreover, the influence of these nanoparticles on the expression of *icaADBC* operon genes and the *icaR* gene was determined. The aim of the study was to determine the influence of silver nanoparticles with a diameter of 10 nm at different concentrations on the ability to create biofilm in *S. epidermidis* bacteria.

## 2. Results

### 2.1. Determination of Antibiofilm Activity of AgNPs

Similar to the research of our team, Lok et al. found the highest activity of nanosilver with a particle size of 10 nm [5,28]. For this reason, nanoparticles with a diameter of 10 nm at concentrations of 1 µg/mL, 3 µg/mL and 5 µg/mL were used. The Figure below shows the microtiter plate immediately after staining the biofilm (Figure 3). The *S. epidermidis* ATCC 12228 strain was the negative control. In the case of the *S. epidermidis* ATCC 35983 strain, a decrease in absorbance was observed at a concentration of 3 µg/mL. Furthermore, for the *S. epidermidis* ATCC 35984 strain, there was a significant decrease in absorbance at a concentration of 3 µg/mL.

After the biofilm was stained, the absorbance of the microtiter plate was measured. The obtained values are shown in the graph below (Figure 4).

### 2.2. Determination of Viability of Bacterial Cells by Microbial Viability Assay Kit-WST

As part of this research, the viability of *S. epidermidis* bacterial cells was also determined at nanosilver concentrations of 1, 3 and 5 µg/mL. A plot of the absorbance versus the concentration used is shown below (Figure 5).

Additionally, a photo of the microtiter plate was taken after adding the reagent. Below is a photo of the plate (Figure 6).

When analyzing the capacity of the three tested strains to form biofilm, a lack of such capacity was observed for the *S. epidermidis* ATCC 12228 strain (the negative control). On the other hand, the *S. epidermidis* ATCC 35983 strain showed a moderate capacity to form biofilm and the *S. epidermidis* ATCC 35984 strain showed a strong capacity to form biofilm. These dependencies were confirmed and are reflected in Figure 4 (nanosilver concentration of 0 µg/mL). For both strains with biofilm forming capacity (*S. epidermidis* ATCC 35983 and *S. epidermidis* ATCC 35984), a clear decrease in the measured absorbance was observed at concentrations of 3 and 5 µg/mL (Figure 3). At these concentrations, the bacteria are not viable in the presence of nanosilver and do not form biofilm.

The viability of the tested microorganisms in the same nanosilver concentrations was slightly different. A significant decrease in absorbance occurred at a nanosilver concentration of 5 µg/mL for all tested strains. Taking into account only the biofilm strains, there was a slight increase in viability at a nanosilver concentration of 3 µg/mL. Figure 4 and Figure 5 suggest that nanosilver at a concentration of 3 µg/mL eradicates the bacterial biofilm and nanosilver at a concentration of 5 µg/mL has a stronger antimicrobial effect on the released cells.

### 2.3. Determination of Gene Expression by Real Time PCR

After Real Time PCR was performed, the test samples were electrophoresed to confirm the specificity of the reaction. Below is a photo of the gel (Figure 7). The visible bands correspond to products with the appropriate number of base pairs (see Table 1).

The results obtained for the four genes tested (*icaA*, *icaB*, *icaC*, *icaR*), with division into individual strains of *S. epidermidis*, are presented below in Figure 8A–D, Figure 9A–D, Figure 10A–D and Figure 11A–D.

The graphs presented above show high expression of all the studied genes in the *S. epidermidis* ATCC 35984 strain. The *S. epidermidis* ATCC 35983 strain showed an intermediate ability to express these genes, whereas the *S. epidermidis* ATCC 12228 strain showed no expression of the genes mentioned, hence it was a negative control. Statistical significance was demonstrated for *S. epidermidis* ATCC 12228 and *S. epidermidis* ATCC 35984 strains in most cases.

For further analysis, the obtained results were divided according to the obtained expression levels, depending on the concentration of nanosilver used. For the expression of genes at individual concentrations, a ratio was calculated. We assumed a ratio of 1.000 for gene expression without the use of nanosilver (Figure 12).

When comparing two biofilm strains (*S. epidermidis* ATCC 35984 and *S. epidermidis* ATCC 35983), statistically significant differences in the expression of individual genes at different nanosilver concentrations for each strain were demonstrated (Figure 12). In the case of the *S. epidermidis* ATCC 35983 strain, a statistically significant increase in expression was observed for all the tested genes at a nanosilver concentration of 3 µg/mL, compared to the negative control (without the addition of nanosilver). Such a phenomenon could act as a defense mechanism against nanosilver, which is an unfavorable factor of the external environment. At the same time, for the *icaR* gene, the observed increase in expression was the smallest when compared to other genes at a nanosilver concentration of 3 µg/mL. At a nanosilver concentration of 5 µg/mL, the expression of these genes returned to the level of untreated bacteria or was even lower. Additionally, in the case of this strain, expression of the *icaA*, *icaB* and *icaC* genes for each tested concentration of nanosilver (1 µg/mL, 3 µg/mL, 5 µg/mL) was higher than expression of the *icaR* gene, compared to expression without the addition of nanosilver. It is also interesting that expression levels of the *icaR* gene only at nanosilver concentrations of 1 µg/mL and 5 µg/mL fall below the expression levels for the case without the addition of nanosilver.

The situation is different for the *S. epidermidis* ATCC 35984 strain. For the *icaA*, *icaB* and *icaC* genes, the highest increase in expression was observed at a concentration of 1 µg/mL compared to expression without the use of nanosilver. In the case of these genes, expression decreased at a nanosilver concentration of 3 µg/mL, whereas at a concentration of 5 µg/mL, it was below the level of expression without the use of nanosilver. For the *icaR* gene, an increase in expression was already present at a nanosilver concentration of 1 µg/mL, and was even more so at a concentration of 3 µg/mL. At a concentration of 5 µg/mL, similar to the rest of the genes tested, the expression level dropped below the level of expression observed without the use of nanosilver. Further, this strain shows a clear difference in the increased ratio for the *icaR* gene (a repressor) with respect to the increased ratios for the *icaA*, *icaB* and *icaC* genes at nanosilver concentrations of 1 µg/mL and 3 µg/mL, suggesting *icaR*-dependent expression for the latter three genes.

The presented results show the influence of nanosilver in different concentrations on the expression of genes of the *icaADBC* operon and the *icaR* gene. The difference in the expression of these genes, depending on the tested strain and its ability to produce biofilm, has also been proven. For the *S. epidermidis* ATCC 12228 strain, expression of the mentioned genes did not occur—it was a negative control. In turn, expression of these genes in the other two strains showed different trends depending on the strain. For most genes, there was an increase in expression at the first low concentration of nanosilver used (1 µg/mL) and a complete decline in their expression to or below the expression level of untreated cells at the highest nanosilver concentration used (5 µg/mL).

## 3. Discussion

The control of staphylococcal infections, especially with their ability to form biofilm, is a serious problem in therapy. Among the alternative substances, the anti-biofilm effect of nanosilver has so far been proven. It can be used to prevent *S. epidermidis* infections, especially those associated with the implantation of biomaterials, vascular catheters or prostheses. In this study, the effect of nanosilver (with a particle size of 10 nm at concentrations of 1 µg/mL, 3 µg/mL and 5 µg/mL) on the viability and the ability of *S. epidermidis* bacteria to produce biofilm. Platania et al. investigated the activity of silver nanoparticles (size <10 nm) in relation to the *S. epidermidis* strain. This team observed no growth at a concentration of 3.1 µg/mL up to 24 h. In this study, viability was maintained at a concentration of 3 µg/mL, whereas viability was significantly reduced at a concentration of 5 µg/mL [22]. In a study conducted by Ni et al. a 30% decrease in growth was observed after 24 h of *S. epidermidis* incubation with silver nanoparticles at a concentration of 5 µg/mL and with a particle size of 10–25 nm [29].

The expression of the biofilm-forming genes belonging to the *icaADBC* operon and the *icaR* gene varied between strains with biofilm producing capacity. When comparing the expression of individual genes at a nanosilver concentration of 0 µg/mL, statistically significant differences were noticed for the *S. epidermidis* ATCC 35983 and *S. epidermidis* ATCC 35984 strains, substantiating differences in the biofilm producing capacities of these strains. At nanosilver concentrations of 1 µg/mL and 3 µg/mL, expression of the *icaR* gene for both tested strains was increased, which could be induced by the activation of mechanisms inhibiting the production of biofilm. This was also confirmed in the phenotypic study (Figure 4). A statistically significant decrease was observed in the expression of the *icaR* gene at a concentration of 5 µg/mL compared to a concentration of 3 µg/mL for both strains. This may be caused by the repressor effect of the gene at a concentration of 3 µg/mL and the lack of biofilm formation at a concentration of 5 µg/mL. It is also noteworthy that this study verified the effect of nanosilver on the ability to form biofilm, not on the mature structure.

The obtained results may have implications concerning the use of nanosilver preparations in antibacterial therapy. The observed phenomena indicate adequacy of the use of nanosilver in sufficiently high concentrations in vivo. The lack of a sufficiently high concentration, or its decrease during the application of the preparation, increases the expression of genes, increases their ability to create biofilm and may lead to increased production of this structure by bacterial cells under such conditions. An example of such a phenomenon is the use of 1 µg/mL nanosilver concentration. For both *S. epidermidis* ATCC 35983 and *S. epidermidis* ATCC 35984 strains, there was an increase in the expression of the *icaA*, *icaB* and *icaC* genes in this case. For the *S. epidermidis* ATCC 35983 strain, there was an even greater increase in the expression of the genes mentioned at a nanosilver concentration of 3 µg/mL. At the appropriate concentration of nanosilver (5 µg/mL) used with these biofilm strains, gene expression drops below the level of untreated cells. Similar results were obtained by Liu et al. with the use of surfactin for *S. aureus*. Expression of the *icaB* gene increased during the application of this substance [30]. The performed experiments indicate that the role of the *icaR* gene in inhibiting biofilm formation is also important. As reported in the work of Morales-Laverde et al., nucleotide polymorphism in the *icaR* coding region leads to an increase in *icaADBC* transcription and PIA production [31]. Similar results were reported by Wang et al., who demonstrated an increase in the expression of the *icaR* gene and a decrease in the expression of the *icaA* gene with the use of silver nanoparticles [32]. As in the study conducted by Benthien et al. who studied the effect of deletion of the *spoVG* gene on the expression of the genes of the *icaADBC* operon, for the *S. epidermidis* ATCC 35984 strain, nanosilver at concentrations of 1 µg/mL and 3 µg/mL may reduce *icaADBC* transcription by modulating *icaR* transcription [33]. On the other hand, in the case of the *S. epidermidis* ATCC 35983 strain, lower *icaR* gene expression compared to the *icaADBC* operon gene expression at a nanosilver concentration of 3 µg/mL may be due to the fact that *tcaR* plays the role of the repressor in this situation. Such a phenomenon was confirmed by Hoang et al. and this implies that the main repressor is *icaR.* TcaR and IcaR may have overlapping binding sites in the *icaR-icaA* intergenic region. Interestingly, this work also noted that TcaR could inhibit *icaR* transcription [34]. Therefore, we conclude that nanosilver reduced biofilm formation and increased the susceptibility of bacteria to silver nanoparticles. It also changed the transcription of biofilm-related genes, depending on the concentration used.

## 4. Experimental Section

### 4.1. Bacterial Strains, Media and Reagents

The second passage of reference *S. epidermidis* strains were used in this study. The first strain of *S. epidermidis* ATCC 12228 (negative control) did not have the phenotypic ability to produce biofilm. The *S. epidermidis* ATCC 35983 strain had moderate biofilm production capacity and possessed the genes of operon *icaADBC*. The *S. epidermidis* ATCC 35984 strain had substantial biofilm-generating ability and possessed the *icaADBC* operon genes. All three strains were obtained from the American Type Culture Collection. Bacterial strains were stored in Tryptic Soy Broth (TSB) medium with the addition of 20% glycerol at a temperature of −86 °C. Silver nanoparticles with a particle size of 10 nm (TEM) at a concentration of 20 µg/mL in aqueous buffer and containing sodium citrate as stabilizer (Sigma-Aldrich/Merck, Darmstadt, Germany) were used in the study.

### 4.2. Determination of the Antibiofilm Activity of AgNPs

The examinations of antibiofilm activity of AgNPs were carried out in accordance with the modified Christensen method [35]. For this purpose, after 18 h (hour) of incubation with silver nanoparticles, the wells were washed three times with phosphate buffer (PBS, pH = 7.2) and all the planktonic forms of bacteria were removed. The plates were dried at a temperature of 37 °C, after which 200 μL of 1% crystal violet (MERC/Sigma-Aldrich, Darmstadt, Germany) was added to each well in order to stain the biofilm. The plates were incubated for 10 min at room temperature. The stain was then removed, plates were flushed four times with deionized water and dried at a temperature of 37 °C. The biofilm that was formed was dissolved in 100 μL of 95% isopropanol in 1 M of HCl. After 10 min incubation, the absorbance was measured at a wavelength of λ = 570 nm, by means of a Multscan EX (Thermo Electron Corporation, Shanghai, China) microplate reader. The assessment of BFA was performed by comparing these absorbance values with those of the *S. epidermidis* ATCC 12228 strain (negative control). The examinations were performed three times.

### 4.3. Determination of the Viability of Bacterial Cells by Microbial Viability Assay Kit-WST

The analysis of bacterial cell viability was carried out using a Microbial Viability Assay Kit-WST, according to the manufacturer’s instructions. We added 10 µL of the Coloring reagent to the bacterial culture with silver nanoparticles after 18 h of incubation. The prepared microplate was incubated at room temperature. After incubation, absorbance of the microplate was measured at a wavelength of λ = 450 nm, by means of a Multiscan EX (Thermo Electron Corporation, Shanghai, China) microplate reader [36].

### 4.4. Determination of Gene Expression by Real Time PCR

#### 4.4.1. RNA Isolation

The bacterial culture with nanosilver was centrifuged (5 min., 6000 RCF) after 18 h of incubation. After centrifugation, the supernatant was removed and 0.8 mL of Fenozol (A&A Biotechnology, Gdynia, Poland) was added to the pellet. The mixture was transferred to a bead homogenizer (Bertin Technologies SAS, Montigny-le-Bretonneux, France) and was homogenized three times (30 s, 2800 RCF). Then, during the next stages, the manufacturer’s guidelines were followed.

#### 4.4.2. Reverse Transcription

The reverse transcription reaction was performed using the Omnisript Reverse Transcription Kit (Qiagen, Hilden, Germany). For reverse transcription, 10× RT buffer, dNTP mix, Oligo-dT primer (EURx Ltd., Gdańsk, Poland), Ribonuclease Inhibitor (EURx Ltd., Gdańsk, Poland) and the Omniscript Reverse Transcriptase were added to the tube, according to the manufacturer’s instructions. Finally, template RNA was added to the prepared mix. Then, a reverse transcription reaction was performed and the tubes were incubated at 37 °C for 60 min.

#### 4.4.3. Real Time PCR Reaction

The Real Time PCR reaction was performed to detect four genes related to biofilm formation capacity: *icaA*, *icaB*, *icaC*, *icaR* and the *gyrB* reference gene. For this purpose, the reagent SsoAdvanced Universal SYBR Green Supermix (Bio-Rad) was used. We added 5 µL of SsoAdvanced Universal SYBR Green Supermix (Bio-Rad), 0.3 µL of each primer, 1.4 µL of water and 3 µL of diluted cDNA template to the reaction. The total reaction volume was 10 µL. The test was performed using the CFX96 Touch Real-Time PCR Detection System (Bio-Rad). The primer sequences are shown below in Table 1. PCR amplification was performed according to the manufacturer’s protocol: pre-incubation occurred at 98 °C for 30 s, followed by 40 amplification cycles with denaturation at 95 °C for 10 s, annealing at 60 °C for 30 s and final melting from 65 °C to 95 °C with an increase of 0.5 °C every 5 s [37].

The results are presented as expression levels (Figure 8) and ratios (Figure 9). The ratio index was calculated by dividing the dCT for a gene tested at a given concentration by the dCT for this gene at a nanosilver concentration of 0 µg/mL. The expression level was defined on the basis of the threshold cycle. The relative expression of genes was analyzed with the use of the relative quantification method 2^(−dCT), after prior normalization with regards to the reference gene [38,39]. The expression of the analyzed genes was normalized with regards to the expression level of the reference gene and the endogenous control gene (*gyrB*).

### 4.5. Statistical Analysis

All experiments were performed in triplicate. Statistical analyses were performed with the use of Statistica 12.5 software. The Shapiro–Wilk test was used to assess the distribution normality. Subsequently, a Kruskal–Wallis test was performed. A *p* value of <0.05 was considered significant.

## 5. Conclusions

In conclusion, our study demonstrates that the antibiofilm effect of nanosilver particles varies. The viability of *S. epidermidis* strains decreases at a nanosilver concentration of 5 µg/dL, whereas these strains have the ability to form biofilm at a concentration of 3 µg/dL. The expression of biofilm-forming genes differs between the individual *S. epidermidis* strains with the ability to produce biofilm. The *S. epidermidis* strains producing biofilm show differences in the expression of genes that determine its formation. The use of nanosilver in subinhibitory concentrations leads to an increased expression of some genes of the *icaADBC* operon. Low concentrations of nanosilver may cause increased biofilm production, however no such effect is observed at high concentrations. This confirms that the use of an appropriately high dose of nanoparticles in any future therapy is of utmost importance.

## Figures and Tables

**Figure 1 ijms-23-09257-f001:**
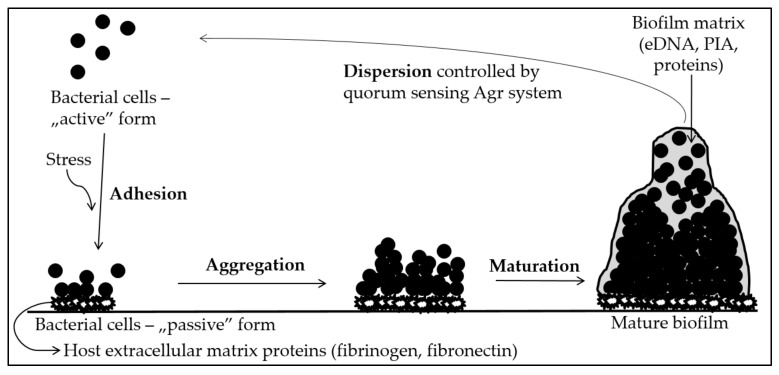
Stages of biofilm formation in staphylococci.

**Figure 2 ijms-23-09257-f002:**
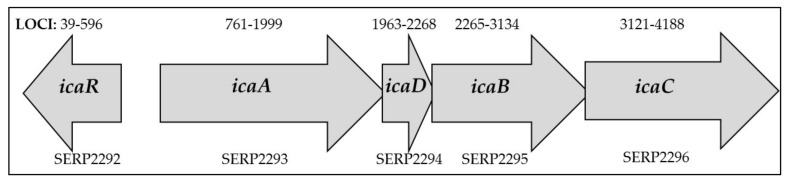
The arrangement of the *ica* operon in *S. epidermidis* with ordered locus names [11].

**Figure 3 ijms-23-09257-f003:**
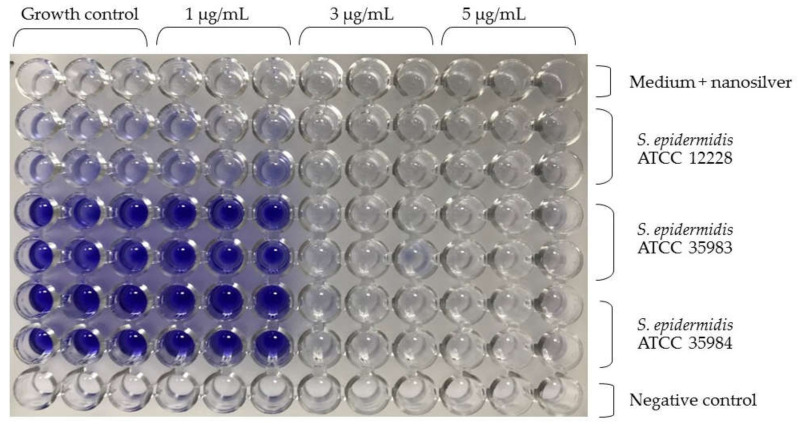
*S. epidermidis* biofilm formation at different nanosilver concentrations. In the upper part there is a description for nanosilver concentrations in triplicate (1 µg/mL, 3 µg/mL, 5 µg/mL). The side part provides a description for the *S. epidermidis* strains tested in duplicate.

**Figure 4 ijms-23-09257-f004:**
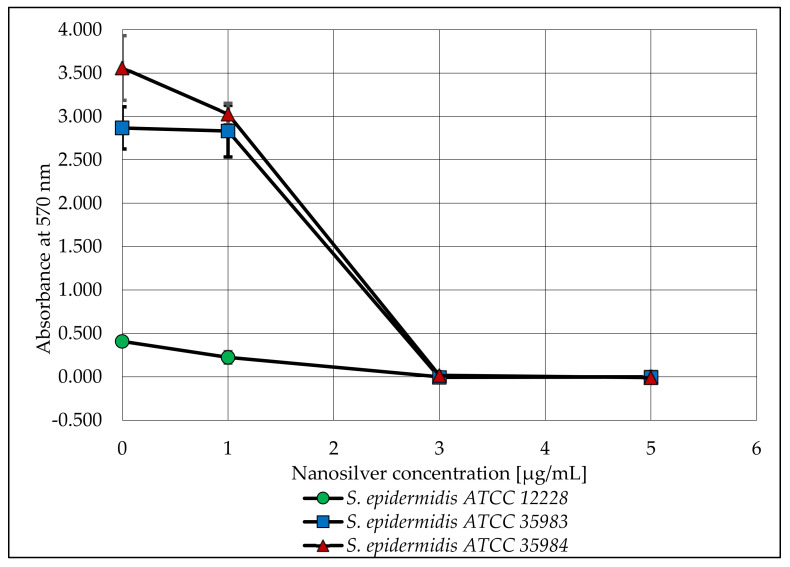
The ability of *S. epidermidis* strains to produce biofilm after exposure to various concentrations of AgNPs with a particle size of 10 nm. Error bars indicate standard deviations.

**Figure 5 ijms-23-09257-f005:**
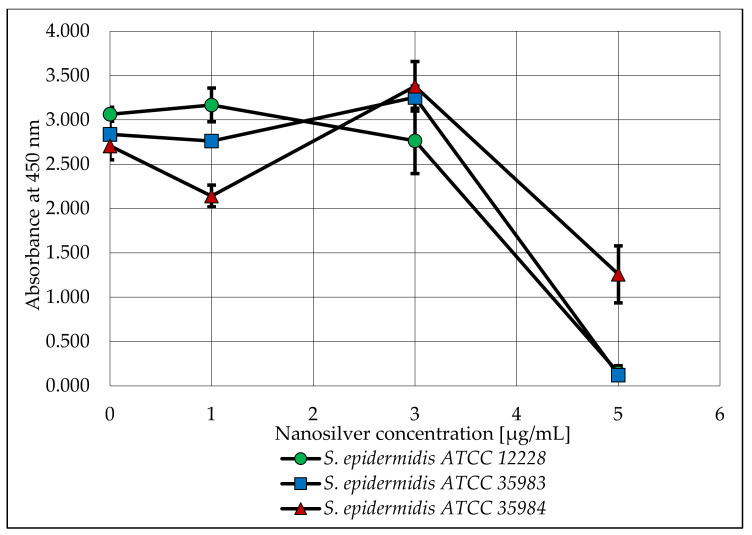
Analysis of the viability of *S. epidermidis* bacteria, depending on the concentration of nanosilver used. Error bars indicate standard deviations.

**Figure 6 ijms-23-09257-f006:**
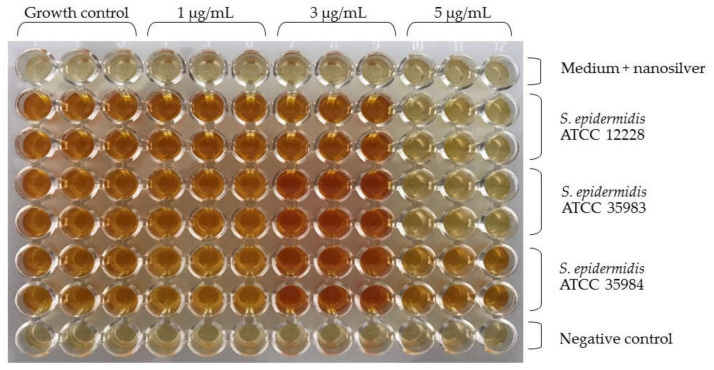
Color change in the microtiter plate as an indicator of decreased *S. epidermidis* viability. In the upper part there is a description for nanosilver concentrations in triplicate (1 µg/mL, 3 µg/mL, 5 µg/mL). The side part provides a description for the *S. epidermidis* strains tested in duplicate.

**Figure 7 ijms-23-09257-f007:**
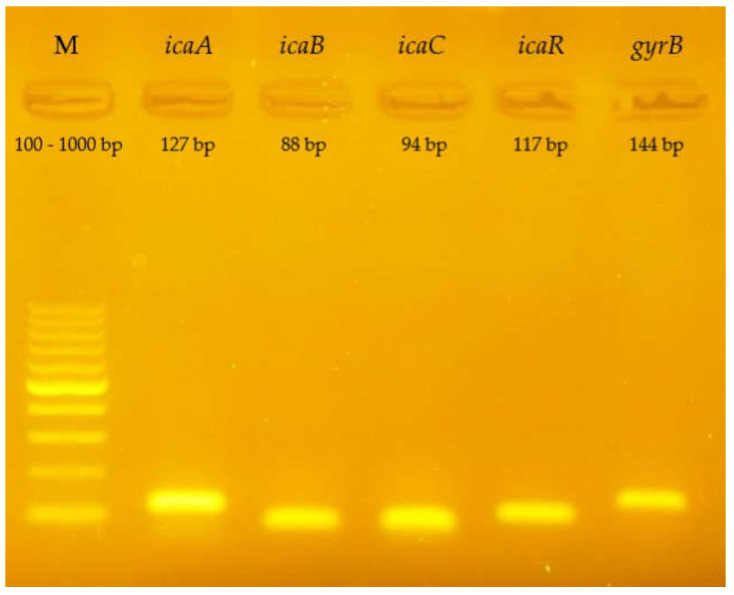
Visualization of the gene amplification products of the *icaADBC* operon and the *icaR* gene (*S. epidermidis* ATCC 35984) in a 2% agarose gel stained with SimpleSafe EurX, Poland; *icaA*—127 bp, *icaB*—88 bp, *icaC*—94 bp, *icaR*—117 bp, *gyrB*—144 bp; bp—base pairs; M—marker.

**Figure 8 ijms-23-09257-f008:**
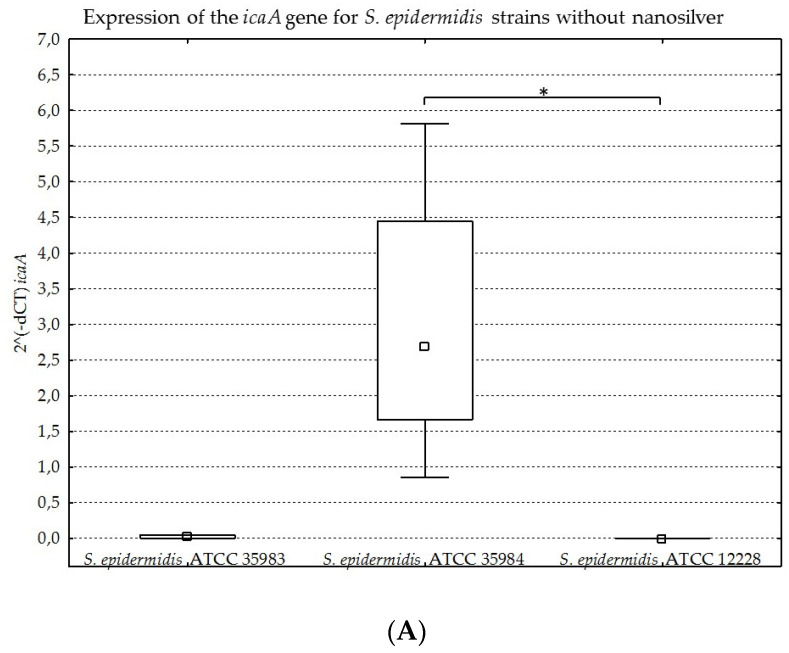
(**A**–**D**). Gene expression (*icaA*, *icaB*, *icaC*, *icaR*) depending on the strain of *S. epidermidis* without nanosilver, presented as medians and quartiles (Q1 and Q3). Statistical significance: * *p* < 0.05.

**Figure 9 ijms-23-09257-f009:**
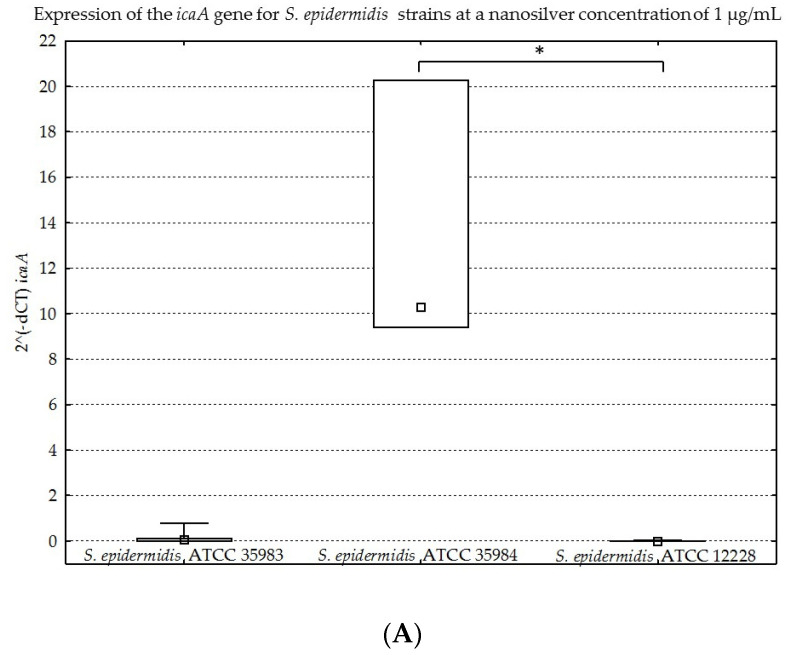
(**A**–**D**). Gene expression (*icaA*, *icaB*, *icaC*, *icaR*) depending on the strain of *S. epidermidis* at a nanosilver concentration of 1 µg/mL, presented as medians and quartiles (Q1 and Q3). Statistical significance: * *p* < 0.05.

**Figure 10 ijms-23-09257-f010:**
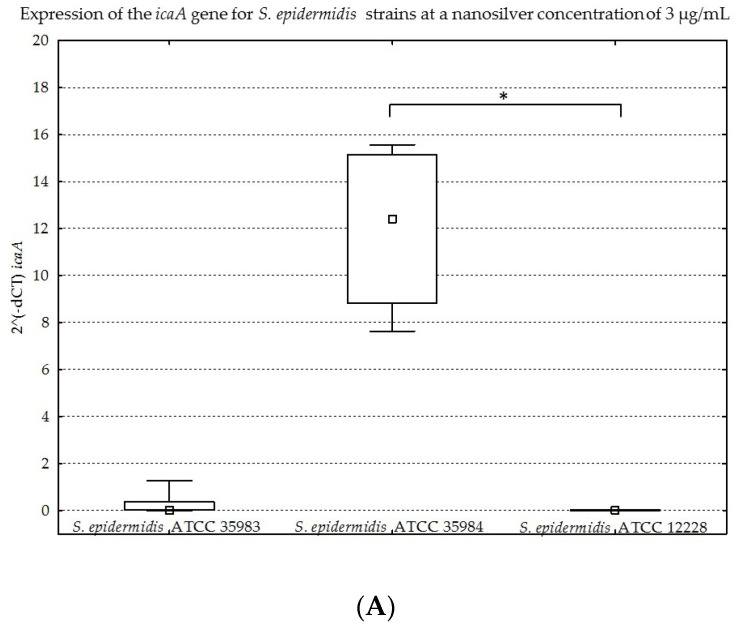
(**A**–**D**). Gene expression (*icaA*, *icaB*, *icaC*, *icaR*) depending on the strain of *S. epidermidis* at a nanosilver concentration of 3 µg/mL, presented as medians and quartiles (Q1 and Q3). Statistical significance: * *p* < 0.05.

**Figure 11 ijms-23-09257-f011:**
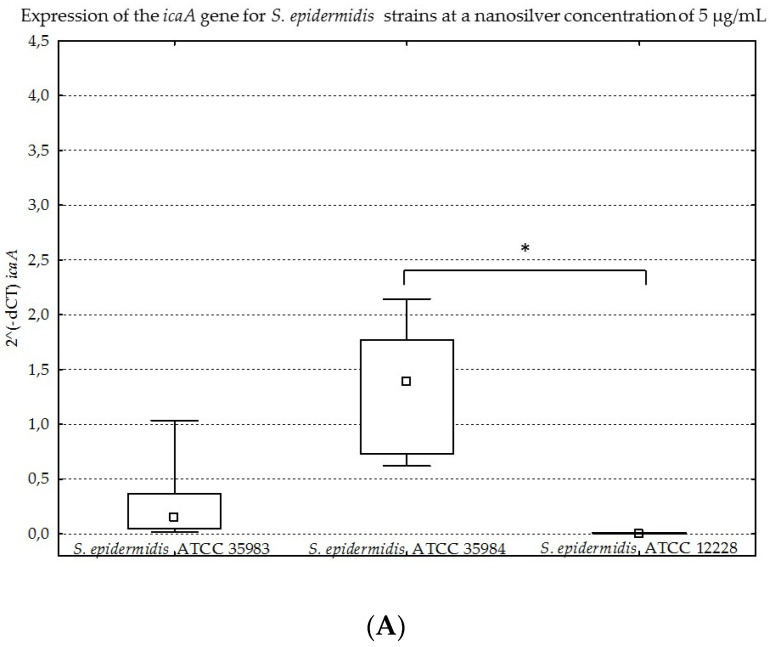
(**A**–**D**). Gene expression (*icaA*, *icaB*, *icaC*, *icaR*) depending on the strain of *S. epidermidis* at a nanosilver concentration of 5 µg/mL, presented as medians and quartiles (Q1 and Q3). Statistical significance: * *p* < 0.05.

**Figure 12 ijms-23-09257-f012:**
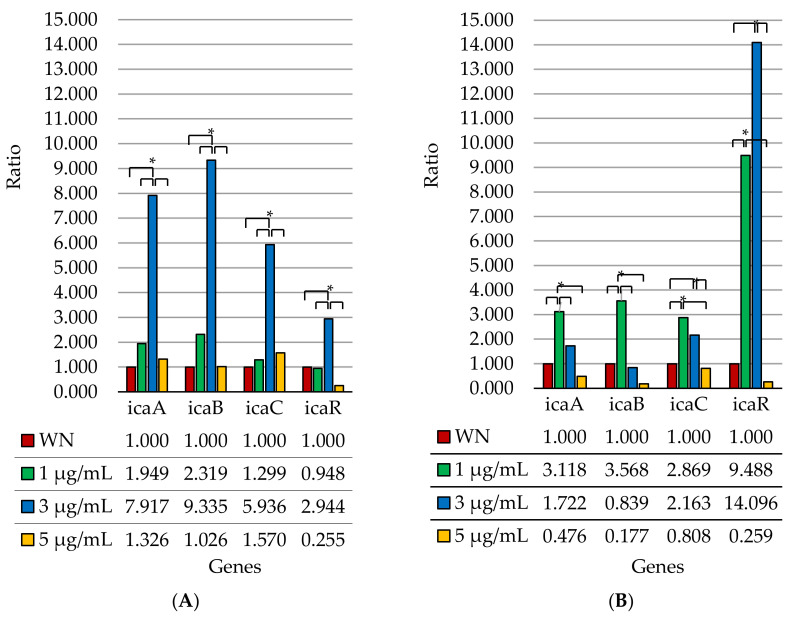
Expression of the genes of the *icaADBC* operon and the *icaR* gene as a ratio index, depending on the concentration of nanosilver used for the *S. epidermidis* ATCC 35983 strain (**A**) and for the *S. epidermidis* ATCC 35984 strain (**B**); WN—without nanosilver; statistical significance: * *p* < 0.05.

**Table 1 ijms-23-09257-t001:** Sequences of primers used in the RT-qPCR reaction.

Gene	Oligonucleotide Sequence	Amplimer Length [bp]	Ref.
*icaA*	Forward: 5′TGGTTGTATCAAGCGAAGTCA3′ Reverse: 5′ATCCTCAGTAATCATGTCAGTATCC3′	127	This study
*icaB*	Forward: 5′CTGTCACACCAGATGCCGATAACTA3′ Reverse: 5′CCGTCCCATTCCTTTATTAGCGTTTC3′	88	This study
*icaC*	Forward: 5′GGCGTCGGAATGATGTTAAGAGA3′ Reverse: 5′AGTTAGGCTGGTATTGGTCAAATTGT3′	94	This study
*icaR*	Forward: 5′GCGATGTGCGTAGGATCATAA3′ Reverse: 5′TGTTCAATTATCTAGTGCTCCAGAAG3′	117	This study
*gyrB*	Forward: 5′TGGTGCTGGACAGATACAAGT3′ Reverse: 5′CCTGCTAATGCCTCGTCAATAC3′	144	This study

bp—base pairs.

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
