# Peer review of "Antibiofilm Effect of Silver Nanoparticles in Changing the Biofilm-Related Gene Expression of Staphylococcus epidermidis"

_ijms, 2022, doi:10.3390/ijms23169257_

Round 1
Reviewer 1 Report
First, please carefully read the whole manuscript to check the quality of the english language and style.
Specific points concerning the content:
The Title (lines 1-4) is appealing. It creates the expectation in the reader that the study will put in evidence a correlation between biofilm formation inhibition and changes in the biofilm-related genes expression. It is a pity that this happens only partially because the presented data is poorly discussed.
The Abstract (lines 12-22) is ok.
The Introduction (lines 25-136) is too long and redundant, with respect to the experimental part. In my opinion, there is no need to describe in such detail the process of biofilm formation. For instance, Figure 1 is not necessary because it could be found in many recent reviews dealing with this topic. This part (lines 36-57) could be reduced or even deleted. A similar consideration applies to the part describing the ica operon-independent biofilm formation (lines 93-108), and biofilm dispersion (lines 109-114), as there is no data about these aspects in the manuscript. I would suggest to keep the focus on the PIA adhesin (from line 58 onward) encoded by genes of the ica operon. Moreover and most importantly, the role of the icaR protein should be better explained (lines 84 – 91).
Results and Discussion (line 137)
I guess that, given the importance of the Discussion section, you might want to separate it from the presentation of results. The title of this section should then be “Results,” followed by the Discussion section.
Lines 139-143: This part is redundant, it repeats concepts already expressed above in the Introduction (lines 131-136). I would suggest to move it to the Introduction, provided the repetitions are avoided.
Lines 194-197: These two statements refer to the statistical analysis of data presented in Figure 8. The statistical significance is not clear enough neither in the figure, nor in the text.
Lines 207-215: In this part of the Results section, the authors focus the attention to statistically significant differences in the expression data displayed in Figure 9. In addition, in the middle of this paragraph, they jump again to something displayed in Figure 8 (lines 209-211). However, it is not clear whether such differences are between the two strains (ATCC 35984 and 35983, as stated on line 207), or between the different nanosilver concentrations for the same strain (as stated on line 208-209). The latter seems more reasonable but, again, the analysis was made with respect to what? I guess, with respect to the expression without nanosilver. In my opinion, a more clear representation of this analysis in figure 9 would improve the understanding of this part of the text.
Graphs and Figures.
Figure 3 and Figure 6 could be removed or at least moved to Supplementary material. Nowadays there is no need to show such standard procedures. To show the results, the quantification reported in Figure 4 and 5 is better. In case the authors should decide to keep the Figs 3 and 6, they should amend them, as in both these figures the top labels are completely misaligned. In addition, the readings reported in Figures 4 and 5 reach very high values, especially those in Figure 5. I wonder about the linearity of such readings. Did the authors make a titration curve in order to assess the linearity of the absorbance in such high range of values? This would be very important to understand whether the variations are significant or not.
Concerning Figure 8, all the labels’ fonts are too small, making them very difficult to read. Please, correct them. Most importantly, the statistical significance between (or among) different strains is difficult to interpret. The authors should display this item in the graphs more clearly, perhaps with a square bracket connecting the box-and-whisker plots to be compared. Statistical significance in Figure 9 is also not clear enough.
Discussion
Line 216: Please, correct the subtitle “2.4 Discussion”, which should be “Discussion” only.
Lines 224-232: The findings concerning a different biofilm formation ability of the three S. epidermidis strains are not a novelty, so considerations on this point should be kept to a minimum.
Line 232: In my opinion, the statement “It is also visible as a color change on the microtiter plate in Figure 3” should be deleted.
Lines 235-237: The authors claim there was an increase in viability of the biofilm-forming strains, because they observe an increase in absorbance. I already expressed my concern about the linearity of absorbance readings in the range above 2.000 (see my comment on figure 5 above).
Lines 237-240 and lines 240-242: In the first statement the authors suggest that S. epidermidis ATCC 35984 could be protected from the action of nanosilver at 5 mg/ml “by the surrounding biofilm”, whereas in the second they conclude that “nanosilver at 3 mg/ml eradicates the bacterial biofilm”. It is a bit contradictory, as it is not clear whether, at 5 mg/ml nanosilver, the biofilm is present or not. In general, it is strange that a lower concentration eradicates biofilm, and a higher concentration has a “bactericidal” (line 241) action (by the way, to say that a compound is “bactericidal”, one should quantify the bacterial survival, usually by viability counts (i.e. CFU), and demonstrate that survival is less than 0.01%), without knowing what this concentration does to the biofilm. In my experience, various antimicrobial compounds show anti-biofilm activity at concentrations higher (sometimes much higher) to MIC/MBC. Of course, it depends on whether the experiment is performed with an established, mature biofilm (which would require a high concentration of the antimicrobial compound), or with a biofilm developed continuosly in the presence of the antimicrobial compound. In this latter case, which is also the case of the study, it is reasonable to postulate that bacteria are simply not viable in the presence of nanosilver, so they do not form biofilm. The authors made a single end-point analysis of bacteria viability at 18 h incubation. Based on this alone, it is difficult to infer cause-effect correlations.
Lines 244-247: As I already said, statistically significant differences are difficult to understand (see above the comment to Figure 8).
Lines 248-251: The authors comment on the expression of the icaR gene, which was “similarly” (line 249) increased in both biofilm-forming strains at 1 and 3 mg/ml. I don’t see such similarity, as quantitatively such increase is different in the two strains. For the 35983 strain, the increase of icaR gene expression is evident at 3 mg/ml only, whilst for the 35984 strain, the increase is evident at 1 mg/ml, and even more at 3 mg/ml (Figure 9). Further, this latter strain shows a clear difference in the increased ratio of icaR (a repressor) with respect to the increased ratio of the ica A, B, and C genes at 1 and 3 mg/ml, suggesting a somehow icaR-dependent expression for the latter three genes. The situation for the 35983 strain seems completely different in this regard, because the ratio of the ica A, B, and C gene expression is always (i.e. at all three nanosilver concentrations) higher with respect to the icaR ratio (Figure 9). So, I wonder where the authors see a similarity.
Line 254: The definition “responsible for the biofilm production process” could be misleading and generating confusion, being the icaR gene a repressor. By the way, the role of this gene is not explained clearly enough in the Introduction (see above my comment to the same topic).
Lines 256-274: In these two paragraphs the authors repeat the results without a real discussion about the meaning and/or relevance of data.
Lines 275-280: Here the authors make some important considerations about the relevance of in vitro data for in vivo (therapeutic) applications. In the statement “increases the expression of genes”, please, add what genes are increased, and cite a relative reference.
Lines 282-289: Here the authors make the only three citations of the published literature in the Discussion section. In my opinion, in a scientific paper data should be discussed also in light of the published literature, especially when data itself is not strong enough to support conclusions. See next comment.
Lines 289-292: In the last statement the authors claim that nanosilver “by inhibiting the transcription of biofilm-related genes reduced the production of PIA and thus reduced biofilm formation etc”. However, data presented in this manuscript does not clearly support such conclusion. I already said above that it is difficult to make cause-effect correlations, based on a single end-point analysis. A kinetic analysis could improve such correlations, unless there are published observations in the literature that can support the author’s hypothesis.
References
There are three references (reference numbers 3, 5, and 7) wich appear twice in the list (reference numbers 9, 28 and 14, respectively). Please, correct.
Author Response
Dear Reviwer,
We would like to express our gratitude for the detailed and critical reading of the manuscript. We are very grateful to the Reviewer for the very positive comments. The revised manuscript has been addressed accordingly all reviewers suggestions.
Reply to the review in the attachment.

Reviewer 2 Report
The manuscript entitled "The Effect of Silver Nanoparticles to Inhibit Biofilm Formation and Change Expression of Biofilm-Related Genes of Staphylococcus epidermidis" presents the results of studies on the antimicrobial properties of silver nanoparticles. The authors' work is combined with global trends and is of scientific interest. The article is written in good language, the narrative is logical, the illustrations are clear. It is proposed to adopt it after minor changes.
1) Figures 3, 6 - the pointers on the right have shifted slightly.
2) Line 188 - extra italics.
3) Line 127 - it is recommended to add a link to https://doi.org/10.3390/mi12121480.
4) Figures 1, 8, 9 - it may be necessary to shift the figures to the right to align with the text.
5) Line 45. It would be appropriate to clarify that the main quality of the surface here is its adhesion rather than its origin.
6) Lines 78, 155. It would be appropriate to add an empty lines separating the text from the figures.
7) Line 108. It should be added that in addition to food, intercellular signals are transmitted inside the biofilm.
8) In the conclusions, it would be useful to point to the result described in lines 266-276. Low concentrations of nanosilver can cause increased biofilm formation, which can be assessed as a negative antimicrobial effect. Such an effect was not observed at high concentrations, which may indicate how important it is to use a sufficient dosage of nanoparticles in future therapy.
Author Response

(The authors gave the same response as above.)

Reviewer 3 Report
The manuscript entitled "The Effect of Silver Nanoparticles to Inhibit Biofilm Formation and Change Expression of Biofilm-Related Genes of Staphylococcus epidermidis" is interest specially with regard to antibiotic resistance of bacteria, but comments in the attached file should be followed

Author Response

(The authors gave the same response as above.)
